# Characterizing Child–Computer–Parent Interactions during a Computer-Based Coding Game for 5- to 7-Year-Olds

Hoda Ehsan [1,*], Carson Ohland [2] and Monica E. Cardella [3]

[1] The Hill School, Pottstown, PA 19464, USA
[2] NASA Johnson Space Center, Houston, TX 77058, USA
[3] School of Universal Computing, Construction, and Engineering Education, Florida International University, Miami, FL 33199, USA
* Correspondence: hehsan@thehill.org

**Abstract:** In this study we characterize ways that interactions children have with their parents and a coding game can support them in engaging in computational thinking. Taking a qualitative approach, we analyzed the video-recordings of 14 families of 5-to-7-year-old children as they played a computer-based coding game in an engineering and CT exhibit at a small science center. The findings revealed a variety of different types of interactions children had with the coding game and with their parents. We discuss the opportunities these interactions provided for children's engagement in different CT competencies. While aspects of the computer interaction were crucial for children's CT engagement, some interactions did not occur in ways that encouraged children's use of CT. Parent–child interactions played a very important role in enabling the children's computational thinking. Overall, we believe the parent–child and child–computer interactions complemented each other to fully engage children in CT. We provide implications for practitioners and designers who aim to support children's engagement in different CT competencies.

**Keywords:** computational thinking; child–parent interactions; informal education

## 1. Introduction

For decades, educational researchers have suggested that computer science (CS) can help children develop skills and processes relevant for solving everyday problems and problems that they will encounter in their future life's work [1–3]. Papert [1] argued that programming computers can further develop children's cognitive processes and allow them to illustrate their ideas in virtual environments while developing problem-solving skills. In 2006, Jeannette Wing [4] introduced the notion of computational thinking (CT) as a thinking process that draws on concepts fundamental to CS; CT "includes a range of mental tools that reflect the breadth of the field of computer science" ([4], p. 33). She emphasized the fact that CS is not just computer programing and is more than being able to program a computer. Instead, Wing suggests that CT is a way of thinking and a fundamental skill required for everyday activities. Other researchers have since argued that CT should be introduced to children as young as elementary- age and be included in different courses, such as mathematics, biological and physical sciences, and social sciences [5].

Since Wing's seminal work in 2006, we have witnessed an increased global shift towards integrating CT in pre-college education [6], including in countries such as the United States of America (CS for All), Australia (Digital Technologies), the United Kingdom (Computing at School), and Mexico [7]. This shift recognizes a need to prepare children for a future that heavily relies on computing [8], as today's children move from being users of technology to innovators and creators [9]. Researchers and educators have created and/or used several approaches to promote children's computational thinking in different formal and informal settings, such as activities for museums [10], curricula for classroom use [10], curricula for out-of-school programs [11], unplugged activities (with no digital

technologies) [12], card and board games [13], apps and computer games [14], and virtual reality technology [15].

At the same time, educational researchers have been exploring ways children engage in CT and effective interventions to promote and support the development of CT in children [16]. For example, Lye and colleagues [16] reviewed 27 existing CT intervention studies and suggested that learning environments should support the CT engagement and development of learners of all ages through constructionism-based problem-solving activities, encouraged by reflection and scaffolding. They reported that in most of the existing studies at the time, learners' computational thinking was fostered through interventional strategies, such as working in pairs and learning from peers, teachers' scaffolding, and computer scaffolding. However, only one study focused on parents' guidance.

Young children typically spend less than 20% of their waking hours in school environments [17], and more than 80% of their time in out-of-school environments. While we often think of school settings as a primary context for learning, in reality children spend considerable time learning in out-of-school settings. In these out-of-school settings, parents and caregivers play critical roles in supporting children's learning [18]. Parents and other caregivers use a variety of strategies to support STEM learning, including through informal discussions; providing access to books, television shows, and toys; engaging children in hands-on activities; taking children to activities, programs, and camps; taking children to designed environments, such as science centers; and using educational kits and resources [19]. In our own earlier work, in a case study of a homeschooling family, we examined an intervention focused on supporting parents and children in constructing understandings of CT that build on the child's everyday and prior experiences. We asked a homeschool mother to facilitate multiple CT activities for her daughter across different settings. We observed that the mother enacted multiple effective roles as she facilitated different CT activities for her daughter [20].

Meanwhile, many researchers have been closely evaluating existing virtual and physical CT resources and interventions designed for children to explore the features that could potentially help children engage in CT [21–23]. Ehsan, Beebe, and Cardella [21] conducted a content analysis of several apps and reported the apps that can promote CT competencies in children. Yu and Roque [22] examined 30 virtual, physical, and hybrid CT kits in terms of the design features of these kits that could support and enable children's CT engagements. They discussed the features that most of the kits have in common, including having coding blocks, providing an opportunity to program a robot or sprite, and having storylines and narratives. While these are very valuable resources for computing education, more research is needed to provide evidence of ways these features could promote CT in young users. Additionally, the review by Yu and Roque [19] overlooks the importance of the human interactions that contribute to learning.

Even with all the work that has been undertaken, many researchers argue that there is still no consensus for how to include CT in children's learning opportunities [21–25], particularly when it comes to the instructional practices [26]. Additionally, some research has not fully considered the ways that teachers, parents, and other adults contribute to CT learning. Thus, more research is needed to understand the best means to incorporate CT in K–12 education.

*Purpose of the Study*

In this study, we focus on one example of a computer-based coding game designed to support children's CT. Taking a qualitative approach, we aim to investigate the interactions that children have with the coding game and with their parents that result in their engagement in CT competencies. We address a gap in the literature where studies have investigated how children interact with coding games or kits, and other studies have investigated how parents support children's engagement in CT, but there is a paucity of research investigating the computer–child–parent system. The research questions that we seek to answer are:

What types of child–computer interactions result in children's CT engagement as children solve problems as part of a coding game activity?

What types of parent–child interactions result in children's CT engagement as children solve problems as part of a coding game activity?

## 2. Computational Thinking

Computational thinking (CT) is a cognitive process and a systematic way of thinking that encompasses several concepts and analytical competencies that help in problem-solving [4]. As previously mentioned, there has been a growing interest in introducing CT in both formal and informal pre-college learning settings. As a result, many models and frameworks have been developed for understanding, implementing, and assessing CT for children [27–31]. These models vary in the ways they have identified, defined, and/or described constructs of CT. For example, Brennan and Resnick [32] defined CT as having the three distinct categories of computational concepts, computational practices, and computational perspectives. Weintrop and colleagues [33] introduced four major categories for CT, including data practice, modeling and simulation practices, computational problem-solving, and systems thinking, and in all, a total of 22 sub-skills were identified. Shute and colleagues [30] identify six "facets" of CT that are applicable to all ages: decomposition, abstraction, algorithms, debugging, iteration, and generalization. BBC education [31] introduces 11-to-14-year-olds to CT by defining and describing four key competencies: Abstraction, Algorithm, Decomposition, and Pattern Recognition. Similarly, the Australian Curriculum, Assessment and Reporting Authority identified five competencies for CT: Abstraction, Algorithm, Data Analysis, Decomposition, and Simulation. Google for Education's CT model [24], however, associates a greater number of competencies with CT: Abstraction, Algorithm Design, Automation, Data Analysis, Data Collection, Data Representation, Decomposition, Parallelization, Pattern Generalization, Pattern Recognition, and Simulation.

In this study, we define CT using Wing's [4] description, as CT is taking an approach to solving problems drawing on conceptual foundations related to computing. We define and operationalize CT using the INSPIRE CT framework (Appendix A). The INSPIRE CT framework has been used in previous research focusing on CT and young children, and was developed through an extensive literature review and synthesis process [34] that resulted in a set of CT competencies that are conceptually similar to the set of concepts identified and defined by Google for Education, with modifications based on empirical research in different formal and informal learning settings. Given the similarities in our scope, we used this framework in this current study. The INSPIRE framework includes eight competencies: Abstraction, Algorithm and Procedures, Automation, Debugging and Troubleshooting, Problem Decomposition, Parallelization, Patterning, and Use of Data [35–38].

## 3. Theoretical and Conceptual Background

This study investigates how children's engagement in CT competencies is supported by interactions with a coding game and interactions with their parents. As discussed in the previous sections, we consider CT a systematic way of thinking that is practiced when solving well-structured, ill-structured, and real-life problems [39]. While this study is informed by literature on CT, it is grounded in constructivist learning theories, which emphasize interactions.

Social constructivist theory suggests that learning is a social process and learners construct knowledge through interactions with others [40]. Vygotsky [40] believed with the assistance of a more knowledgeable other who can be a parent or a more competent peer, children can be positioned in a Zone of Proximal Development, where they can gain practice accomplishing tasks that they would not be able to on their own, gaining skills and knowledge in the process. Piagetian theory posits that knowledge is constructed through learners' active interactions with developmentally appropriate materials and the environment [41]. Given these constructivist theories, we believe that in order for children

to develop CT competencies, children should be exposed to developmentally appropriate activities that posit meaningful problems to be solved. The activities should provide a space for children to explore different concepts of CT while having active partners (i.e., people who have more knowledge or experience with the concept or activity, such as parents who have more experience with solving problems) that support this exploration. In addition, learners develop CT as they interact with the environment around them and make meaning of these interactions. Below, we further discuss the interactions that support CT.

### 3.1. Child–Computer (Coding Game) Interaction

Since Wing's 2006 article, which highlighted the importance of promoting CT in children, researchers and practitioners have provided different opportunities for children to interact with CT activities and have been exploring ways to promote children's engagement in CT, including through various programming and coding activities used across different formal and informal settings [42]. With the growth of the number and variety of introductory CT, programming, and coding activities, there has also been variation in approaches and "languages", (e.g., block-based vs. text-based), but little is known about the ways these interactions promote children's engagement in CT [43]. Therefore, in this study, we are looking at children's interactions with a coding game that results in their engagement in CT.

Inspired by a study conducted by Weintrop and Wilensky [43], we define computer interactions as interactions with any of the components of the entire system of the computer-based coding game. Systems designed to support children's learning of coding and computational thinking can include virtual and/or physical features. The virtual features include the game interface that captures all of the presentations of representations. The representations are the set of symbols and texts used to present different concepts and convey meaning and messages to the user. The physical components include the screen that holds the interface and other possible components, such as coding buttons. Interactions with any combination of these features can engage learners in learning.

Existing coding and programming systems have been designed with different virtual and physical features. For example, Long and colleagues [44] developed, implemented and studied an interactive tabletop system where the physical features (i.e., coding blocks) were connected to a virtual system that created music and light. In another study of children's engagement in CT, Moore and colleagues [41] used a system with no virtual features. They used sets of child-friendly robot mice that did not have any virtual interface, but had several physical components, including the robot mouse, the coding buttons and tiles, and other physical pieces used to create mazes for the mice to navigate. Similarly, Sullivan and colleagues [45] developed KIBO programming blocks to engage children in computational thinking, where the KIBO system was also designed with no virtual components. On the other hand, many coding systems have only included virtual features. This is particularly common in CT mobile and/or desktop-based apps and programs that are mapped with other hardware [21]. One example is the online coding game called 'The Foos' by CodeSpark [46]. Fagundes [38] and colleagues observed that children engage in multiple CT competencies when coding the Foos characters to solve different problems [38]. Another example of these coding systems with only virtual features is the Math on a Sphere web-based environment, implemented in an exhibit at the Lawrence Hall of Science and at the Fiske Planetarium. Researchers tested the software with 8- to 13-year-old children and found that the software supported the children in engaging in computational thinking [47].

The design of the system determines the actions users can take and the interactions they can have with the system. These actions can include selecting arrows or blocks for coding, fixing the codes, and abstracting the symbols. Any of these actions can result in the developing and practicing of computational thinking competencies. In this study, we focused on a coding game system, Coding for the Critters, which has both virtual and physical features. We examined children's interactions with this system as a whole

and the computer-based coding game system's individual features, along with children's interactions with their parents, to capture the children's engagement in CT competencies.

*3.2. Child–Parent Interaction*

Parents' involvement in their children's development is very important during the children's early years [48], for a variety of reasons. Some researchers note that parents are the main influencers of their children's academic success [49]. Consistently with the notion of scaffolding, which is drawn from social constructivist theories [50], parents can help enhance children's STEM learning while they participate in different activities at home, or while visiting designed settings (e.g., museums), or while participating in family-based camps and programs [51]. During these activities, parents are children's "thinking guide" [18,52] as they play the role of the "more knowledgeable other" [39] and provide scaffolding and support, which can support reasoning and thinking in their children.

One-on-one parent–child interactions are also a vehicle for improving children's scientific reasoning and logical thinking skills. In a study of parent-child interaction in a museum exhibit, Crowley and colleagues [53] found that children who interacted with the exhibit with their parents had more opportunities to build concrete scientific thinking skills than similar peers who interacted with the exhibit without their parents. Similarly, Palmquist and Crowley [54], in a study conducted at a natural history museum, demonstrated that parent–child conversations engaged children in complex disciplinary reasoning and solving problems. In a single case study of a 9-year-old child on the autism spectrum, we observed that the parent's involvement helped the child to scope the engineering design problem and build a solution to the problem [55]. In another study, Svarovsky and colleagues [56] investigated one-on-one conversations between parent–child dyads as they were engaged in engineering design tasks. Their study's findings emphasized the significant role of parents in their children's engineering, thinking, engagement, and agency.

Prior literature supports the theory that parent–child interactions can influence the quality of children's computing and CT experiences. Our previous work has shown that with the help of parents, children are able to demonstrate and practice CT while engaging in an engineering design task [12]. In this study [12], we observed that parents' conversations with their children helped them to break down the problem and engage in problem decomposition. Parents' instructions also helped children to engage in following and creating algorithms and debugging algorithms. Parents also provided confirmation and encouragement as children engaged in abstraction. In a different study by our research team led by Rehmat [57], we identified parental instructional strategies that supported five CT competencies in their children in two museum activities: an unplugged hands-on activity and a high-tech coding activity. We found that instructional strategies, such as questioning, modeling, and motivational conversation could support young learners to engage in CT competencies. We presented ways these strategies are applicable in other learning settings and argued that learning from parental strategies is very important, since parents know their children better than anyone else, even content experts. Similarly, Yu, Bai, and Roque [58] stated that even though the parents who participated in their study reported having limited programming knowledge, they played several important roles that supported their children during different coding games. However, they did not observe parent–child interactions, and reported their findings based on parents' interviews. Moreover, the purpose of their study was not to find ways these roles supported specific programming/coding/CT practices or competencies in children. Thus, in this study, we uniquely aimed to uncover what child–parent interactions support children's engagement in each of the CT competencies from the INSPIRE CT Framework.

## 4. Methods

*4.1. Context*

The interactions studied in this paper were observed at an engineering and computational thinking exhibit in a small science center [36] in the Fall of 2018. The exhibit was

designed specifically for 5- to 7-year-old children. The exhibit designers envisioned that parents and other family members would interact with the exhibit alongside the 5- to 7-year-old children, consistently with recommendations from the literature (e.g., [53]). The exhibit is composed of multiple stations, but in this paper, we only focus on the computer-based coding game. To complete the levels of the game, children need to create sequences of moves for an on-screen robot character to move through mazes. The game players are prompted to use the shortest possible route so the robot can deliver medicine to three animals as quickly as possible (see Figure 1). If a player finishes a level but does not find the shortest route, the level completion screen is displayed but the player is prompted to retry the level if they wish to find the shortest path. The game features five levels and a tutorial that introduces the concepts of providing instructions to the robot using large directional buttons, using the "Go" button to run the sequence of moves, deleting moves, and resetting the level. In later levels, additional instructions about the order in which the animals must receive their medicine is given. Obstacles also block certain paths on some levels, and if the robot ever hits an obstacle, a large word bubble "OOPS!" is displayed, the code is cut short, and the robot returns to its original position. The game also features three informational user interface elements: a log of the sequence of moves the player has input already, a log of the moves the robot takes and its interaction with game elements, and a display showing the types of medicine that have already been delivered in a single run. Figure 1 shows some of the components of the exhibit.

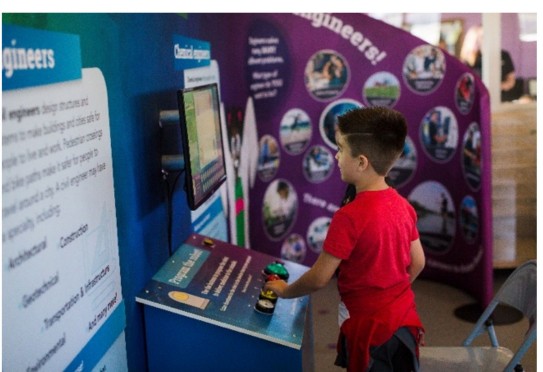
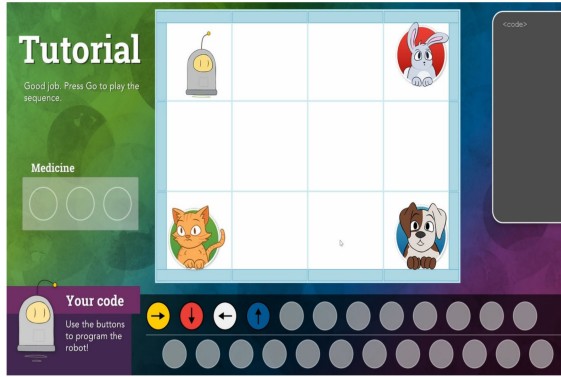
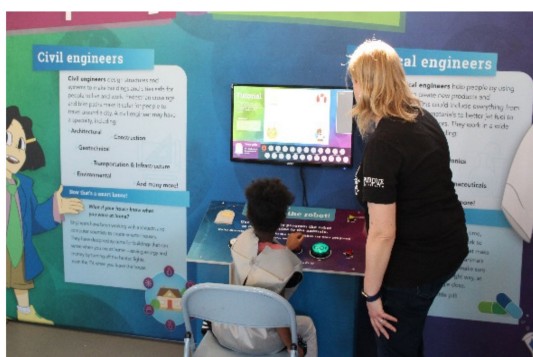
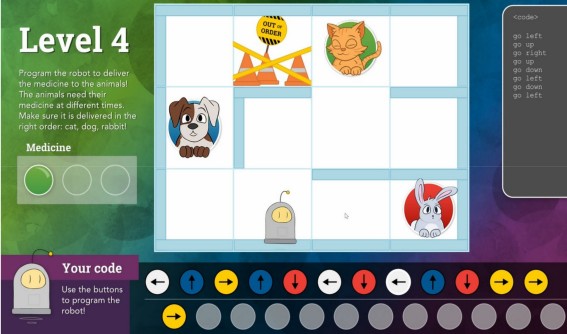

**Figure 1.** Coding game in the exhibit: children must code the robot to deliver medicine to three sick animals.

### 4.2. Participants

The opportunity to participate in the study was advertised to families visiting the science center as well as families of children who previously participated in engineering and computational thinking activities in our partner schools. Fourteen families agreed to participate in this study and signed a consent form that was previously approved by the university's Institutional Review Board. Among the fourteen families, two families were Black, eleven were White and one identified as Multiracial. All families included at least one child who was between the ages of 5–7 (and was enrolled in kindergarten through

second grade). The size of families varied between two (parent and one child) to six (two parents and four children). Although most parents were women, the child participants were equally boys and girls. Families' participation varied between $3\frac{1}{2}$ and 12 min, with some families moving on before completing all five of the levels.

### 4.3. Procedure

The study was conducted at a time when no other visitors were at the science center. This decision was made for two reasons: (1) to collect better quality video and audio data, and (2) to facilitate an in-depth investigation of interactions by limiting potential distractions. All families engaged in the coding game after trying multiple other sections of the exhibit. Families approached the coding game differently. In some cases, children began by exploring the system by themselves and trying different buttons, and realizing the theme of programming a robot to deliver medicine to animals was also part of other sections of the exhibit. In some cases, parents initiated the engagement with the coding game by reading the prompts aloud, encouraging children to guess what the activity was and/or describing the activity in their own words.

### 4.4. Data Analysis

To answer our research questions, we conducted an exploratory qualitative analysis focusing on 5–7-year-olds and their interactions. Their interactions with the exhibit were video- and audio- recorded. We focused on the child's verbal and non-verbal interactions with the coding game and their verbal interactions with their parent(s). To perform an in-depth analysis, we utilized the seven-phase, non-linear video analysis process (Figure 2) suggested by Powell, Francisco, and Maher [59]: (1) viewing the video data attentively, (2) describing the video data, (3) identifying critical events, (4) transcribing, (5) coding, (6) constructing a storyline, and (7) composing a narrative. In phase five, we utilized a three-step coding process (Figure 3), while the video recording was rewatched often as needed. The first step was used as a baseline for capturing all children's engagement in CT, and the second and third steps aimed to answer the two research questions.

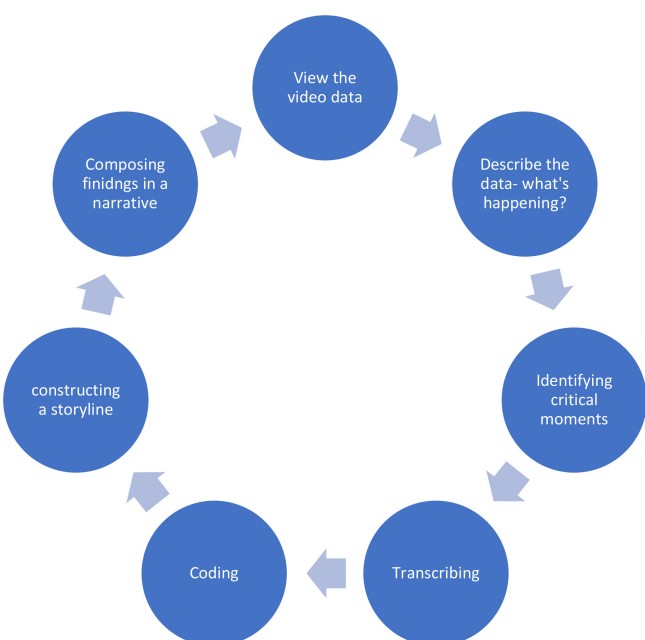

**Figure 2.** Video analysis process adapted from Powell and colleagues (2003).

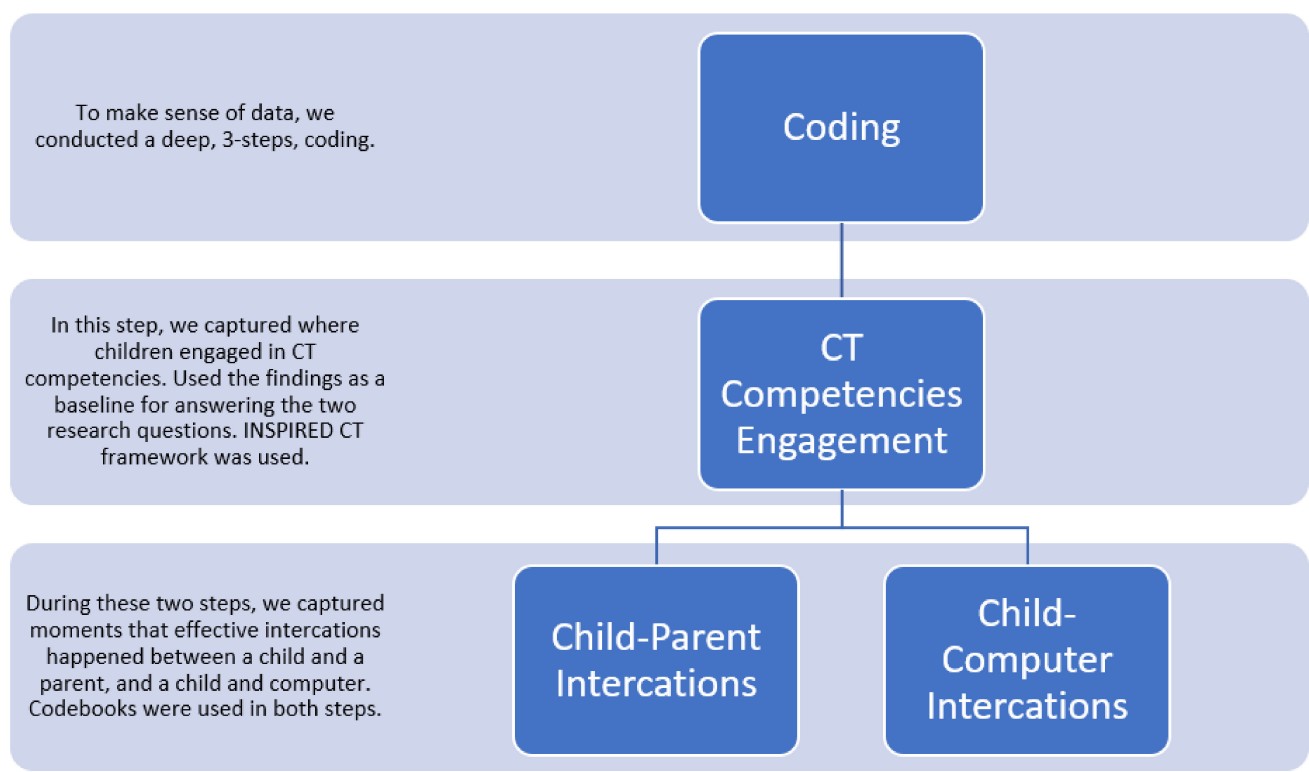

**Figure 3.** Phase5-coding.

(1) Exploring children's engagement in CT. We used the INSPIRE framework for CT competencies as our codebook (see Appendix A). We captured all of the instances of child engagement in CT competencies in this step. We then further examined those instances in the next two steps for possible "effective" interactions. We defined "effective interactions" as any interactions that children had with parents and/or the coding game that resulted in their engagement in CT competencies as identified in the codebook. As a note, the coding game did not provide opportunities for children to engage in all the competencies of the INSPIRE framework. Automation and Parallelization could not happen in this activity.

(2) Exploring effective child–parent interactions. We utilized a child–parent interaction codebook, which was developed based on a synthesis of literature on family/parent engagement with a child learning in informal learning settings. This codebook (see Appendix B) was used in two previous studies [20,60] and was modified based on the findings from those studies.

(3) Exploring effective child–computer interactions. We first closely evaluated the physical and virtual aspects of the coding game, and created a list of all possible interactions the child may have had with the coding game. To confirm these interactions and possibly add to the list, we observed a video of one family (not included in the analysis presented in this paper) and captured examples of these interactions. We then developed the codebook based on the list of possible interactions and examples from one family. This codebook (Table 1) was then used to capture child–computer interactions for video recordings of the families included in this study. In Table 1, we use "N/A" to note that there were some interactions that we did not observe in our study, but that we imagine may have taken place without being observable, or that might have taken place when other families interacted with the exhibit.

**Table 1.** Computer-interaction codebook.

| Interactions | Image from the Exhibit | Description | Examples |
|---|---|---|---|
| Using Buttons |  | The child uses the buttons to enter moves for the robot to follow. | The child begins entering instructions for the robot using the buttons. |
| Reading Written Instructions on the Keyboard |  | The child reads the text on the keyboard or the parent reads the text aloud to the child. | The mother reads the instructions on the keyboard aloud. |
| Reading Written Instructions on the Display |  | The child or a parent reads the instructions on the display (i.e., on the computer screen). | The code fails because the medicine must be delivered in a specific order. The mother prompts the child to read the on-screen instructions. |
| Interacting with the Map |  | The child observes or interacts with the on-screen map. The child may trace different paths, tap out different moves, or simply look at and follow a path in their head. | The child keeps track of the robot's position by placing his finger on the display and following the path through the map. |
| Reading the Code Log |  | The child looks through recorded moves in the code log, to find a mistake or to identify the robot's location at some point in the code. | The child seems to get lost, and the mother tells them to use the code path to see where the robot is at the current point in their path. They follow the moves in the log, but skips a move entered in error earlier. |

**Table 1.** *Cont.*

| Interactions | Image from the Exhibit | Description | Examples |
|---|---|---|---|
| Reading Robot Text-based Code log |  | The child reads back and uses information in the robot's text-based code to identify an error or understand its movement. | N/A |
| Reading Medicine Bar |  | The child uses the medicine bar to understand which of the three animals the robot has already visited. | N/A |
| Receiving Feedback after Running Code |  | The child observes the "OOPS" bubble in a failed path or reads the text displayed at the end of the level. | The robot crashes and the "OOPS" bubble is displayed. The child begins searching for the error in his code. |
| Watching the Robot Move |  | The child closely observes the motion of the robot during a test of the directions they have entered. | N/A |

During phase five, we constantly worked to achieve agreement and validity of our coding. Two of the authors were the main coders. We separately coded six videos in the first round. We then discussed all of the codes until we completely agreed on the codes. In the case of disagreement, we consulted the third author. Throughout this process, we watched and rewatched the videos, and when needed, we revisited the descriptions and made them clearer. When we achieved 100% agreement on all of the codes of the first six videos, the rest of the videos were divided to be coded separately. We present the description and discussion around the findings from the coding in steps two and three in Section 5.

The outcome of phases six and seven are the findings of the study as written in Section 5. In phase six, two authors looked for similarities and differences across all the identified codes and engaged in making meaning of the codes. We looked specifically at the interactions we identified as effective and carefully examined the scene in which those interactions occurred. This examination captured the ways these interactions helped or resulted in children's engagement in CT competencies. Finally, in phase seven, we created the narrative for each interaction, and explained the scenes and how the interactions happened.

## 5. Findings

The purpose of this study was to characterize children's interactions with parents and the coding game that was part of a computational thinking exhibit at a science center,

and the opportunities these interactions provided for children's engagement in different CT competencies. After analyzing the videos of our participants, we observed that both types of interactions provided opportunities for children to engage in CT competencies. While aspects of the computer interaction were crucial for children's CT engagement, some interactions did not occur in ways that elicited children's use of CT. Parent–child interactions played a very important role in enabling children's computational thinking. Overall, we believe both interactions complemented each other to fully engage children in CT. Below we present a description of interactions that resulted in children employing CT competencies.

### 5.1. Child–Computer Interactions

In this section, we present and discuss interactions we observed children having with the coding game.

### 5.1.1. Using Buttons

The exhibit included seven physical buttons that were part of the coding game: left arrow, right arrow, up arrow, down arrow, "Go", "delete", and "reset". The buttons were designed in different shapes and colors (see Figure 1 or the first row of Table 1). They were approximately 1 square inch, and the arrows or text were depicted on top of them. The directional buttons allowed children to enter individual moves for the on-screen robot, and the "Go" button ran the sequence of moves. The delete button removed the last move in the sequence, and if the restart button was held, it reset the current level or the entire game.

Children's interactions with the buttons allowed the children to practice multiple CT competencies, including Abstraction, Algorithm and Procedures, and Debugging/Troubleshooting. We observed that children easily interacted with the buttons given the size, the shape, and the simple information printed on them. Children engaged in an Abstraction process of generalizing the main meaning of the arrows or words printed on the buttons to understand the function of the buttons. When a child interacted with the buttons, they mainly pressed the directional buttons to enter moves for the robot, and pressed the delete and reset buttons to fix errors they found in their existing code. When children interacted with the buttons, they engaged in the Algorithms and Procedures competency. The buttons did not inherently encourage the enactment of this competency, but they did enable this engagement, as children easily used the buttons to create their algorithm (sequence of moves) or follow other's algorithms (created by parents). Likewise, when children engaged in the Debugging and Troubleshooting competency, they had to use either the delete or the reset buttons to erase and rewrite errors in their code. The ease with which the children used the buttons shows that the design of these buttons was developmentally appropriate to engage our target group in CT competencies.

### 5.1.2. Reading Written Instructions on the Keyboard

Instructions introduced the context and objective of the game and were listed above the physical input buttons. Naturally, children interacted with the written instructions when they read them. Alternatively, some parents read the instructions aloud to their children.

The written instructions on the keyboard provided information about the activity that was needed to proceed. Most of the children we observed could not read, and those who could were often too excited about the game to take the time to read them. Even when parents read the instructions aloud, children had frequently already begun trying to enter inputs to the game. We observed only a few instances where families read the instructions before beginning, then referred their children to the text later when they did not get the "best answer". This helped children engage in Debugging and Troubleshooting, as they needed to find paths with fewer moves. However, children in the one family who read the instructions before beginning were able to create more efficient algorithms initially with less need for debugging. Therefore, we believe that if the written instruction were provided in a clear way that gained parents' and children's attention from the beginning, children

could engage in algorithm design in a more effective and efficient way. We suggest that the text could possibly be written in bigger fonts and shorter, simpler sentences.

### 5.1.3. Reading Written Instructions on the Display

Similarly to the keyboard instructions, the instructions on the display contained context for the activity and the game's specific objective. The main difference was that on the first tutorial level, the on-screen instructions displayed specific instructions to teach the player how to move the robot, and on later levels the on-screen instructions provide a specific order in which the robot must reach the different animals. Children could interact with the on-screen instructions by reading them or by listening to their parents read them. However, similarly to the text on the keyboard, the text was not in a large-enough size to gain the children's, or in many cases, the parents' attention at the very beginning.

Since many children could not read the instructions, and many parents either did not notice or did not read the instructions, many families became stuck on later levels involving a particular delivery order. This issue actually created opportunities for parents and children to engage in Debugging. In one narrative, the following interaction occurred: "The younger child enters instructions immediately . . . The code fails because the medicine needs to be delivered in a specific order on this level. After examining the level for some time, the mother re-reads the directions and prompts the children to read them as well to find the medicine order. The child corrects his code and completes the level". Some families were agitated by the tutorial because they did not notice the instructions telling them to use a specific move button. Because they did not use that specific button, they were not able to proceed to the next level. In these cases, the written instructions actually inhibited the ability of these families to further engage in the activity and in the different computational thinking competencies. Those families who read the instructions before beginning the activity, especially those with children capable of reading, were able to decompose the problem more quickly and create effective algorithms. Thus, these children were able to develop and practice CT competencies and were able to complete all of the levels.

### 5.1.4. Interacting with the Map

For each level, a maze-like floor-plan map was shown on-screen. The map was made up of tiles, which were sometimes separated by walls. Some obstacles also appeared that filled an entire tile. Children's interactions with the map were crucial for completing the task. When children looked at the screen at any point during a level, they interacted with the map. The map filled most of the screen and contained almost all of the necessary information to complete the level. Sometimes, children interacted with the map more directly by physically placing their fingers on the screen to trace a path through the level.

Interaction with the map enabled children to engage in Data Collection and Simulation. Children used Data Collection when they counted out the number of moves in different paths, tapping each tile of the path on the on-screen map. Simulating during the activity required the use of the map as well; children placed their fingers on the on-screen map and followed the codes in their code log, simulating the robot's motion using their fingers. Naturally, the map also enabled all other competencies used during the activity, since the map was such an integral part of the activity.

### 5.1.5. Reading Code Log

The code log was a record of entered codes that showed the various moves in different colors. When the "Go" button was pressed, the robot followed all of the movement instructions listed in the code log. Children interacted with the code log when they used their finger to trace the movements in the code log to locate their position or to find an error. This interaction happened either during coding, before pressing GO and testing the code, or after the code was tested and an error was noted.

The code log was primarily used when children practiced Debugging, though it was not always used successfully by children. In one family's interactions, a child hastily

entered inputs for a level and a parent instructed the child to "double check, dude". The child used their finger to simulate the motion of the robot, following the moves recorded in the code log. Eventually, the family identified a missed input and used the code log to edit the move inputs to fix the issue. When children followed along moves without parents' help, they sometimes missed errors by tracing the path they meant to notate instead of identifying a mistake in the recorded path. In some instances, the code log also supported Algorithm and Procedures by helping children to keep track of the moves they entered; one child became confused partway through a level, so they followed the code log up to the most recent input to identify the robot's position at that point in the code. The existence of the code log was very important because young children are unlikely to be able to keep long sequences of moves in their working memory.

### 5.1.6. Reading Robot Text-Based Code Log

The robot text-based code log was a boxed log at the top right of the display that showed the information sent to and the feedback from the robot in a rudimentary programming language. As the move sequence was played, the text-based code log would display each move as "move left", "move right", "move up", or "move down". When the robot collided with an obstacle, the robot text-based code log displayed "hit wall". If children interacted with the code log, they would read it and use it to prompt or aid in debugging. Unfortunately, no child was observed utilizing the code log, though some parents did notice and use the log.

The text in the box was too small and moved so quickly that it almost never provided players with meaningful or usable information. In some cases, parents noticed when the log read "hit wall" after a collision, but most families noticed the displayed "OOPS!" on the screen in a faulty path instead. In the few cases when the robot text-based code log was used by parents, they conveyed the message to their children that there was an error in their code and they needed to fix it. Thus, the log became effective in encouraging Debugging and Troubleshooting.

### 5.1.7. Reading Medicine Bar

The medicine bar was a user interface element displaying which animals had received medicine from the robot during a run of a movement sequence. It featured three empty circles that were filled with their respective animal's color when that animal had had its medicine delivered. If children used the medicine bar, they could identify the different circles that were filled or empty, allowing them to learn which animals the robot had successfully delivered to. Unfortunately, since the information was only briefly displayed during a run of the code, and since the bar was never explained, there was no evidence that any children used the bar and learned what the bar meant.

As with the robot text-based code log, the medicine bar very rarely helped any families in utilizing CT competencies. In a few instances, when the families reached the levels with ordered deliveries, parents noticed that only one bubble was filled in, indicating that they had only delivered to the last animal they reached. Most families, however, never seemed to notice the bar, or at least never realized what information it meant to convey. In the few cases where a parent noticed a single bubble being filled in at the end of a run instead of all three, the bar prompted the parent to read the instructions and realize that a specific order of delivery was required, enabling the children and family to engage in Debugging and Troubleshooting.

### 5.1.8. Receiving Feedback from Running Codes

Children interacted with the screen very often and received feedback as they engaged in coding. If the robot hit a wall or obstacle when the children ran their sequence, a large, jagged word-bubble "OOPS!" was displayed at the robot's position. Additionally, after a level was completed, a text slide was displayed that congratulated the player. If the entered

path was the shortest path, the slide said so. Otherwise, the slide prompted the player to try again to find a path with fewer moves.

One of the most useful and relevant forms of feedback in this coding game was the "OOPS!" bubble displayed when the robot hit a wall. Almost all participants, including children, understood that the robot had collided with an obstacle when the word bubble was displayed. Even children who could not read the word likely understood the meaning due to the iconography of the jagged and brightly colored outline. After the bubble was displayed, almost all participants understood that they needed to use Debugging and Troubleshooting. The "OOPS!" bubble was very effective at encouraging the use of this competency. The feedback given to the player between levels was less effective than the "OOPS!" bubble sign. The two slides, which indicated whether the player should try to improve their path, appeared very similar, and many children could not read the slides in order to realize they should repeat a level if they had not found the optimal solution. In one case, parents read the slide aloud, but the child pressed the "Go" button to move to the next level instead of choosing to go back and find the best answer on the previous level. Such occurrences were common; when children missed the solution with the fewest moves, they usually continued without improving their solution, likely without understanding the text-based feedback given.

### 5.1.9. Watching the Robot Move

When a child entered a sequence of moves and pressed the "Go" button, the robot followed their entered moves exactly, moving through the on-screen maze one tile at a time. Children interacted with the robot's movement by closely following the robot as it maneuvered through the maze along their path. Though some children were excited to see the robot following the path they defined for it, none watched the robot carefully to identify the location of any errors. This was evident due to the fact that we did not observe any instances where children identified the error in the coding sequence by watching the robot moving in the first attempt. The robot moved very quickly along the defined route, so it was difficult to identify its exact point in the code during a run. Occasionally, families ran the simulation a second time to identify the physical location of an error, but generally families only observed the robot moving on a cursory basis to see whether or not it would crash.

### 5.1.10. Combinations of Interactions

While we discussed each type of interaction separately, we need to acknowledge that a combination of these interactions, the system as a whole, resulted in children solving the problem computationally. The components of the coding game were designed to work together to promote the use of CT competencies in children. In one instance, a child entered instructions for the robot using the buttons. She watched the map and kept track of the robot's position using her fingers. She then pressed "GO", and with no errors she completed the level.

Here, the child used multiple computer interactions to engage in the Algorithm competency. She interacted with the map to determine the proper path through the maze and entered her instructions for the robot using the interface buttons.

### 5.2. Parent–Child Interaction

In this section, we present and discuss different parent–child interactions we observed happening while the families interacted with the coding game.

### 5.2.1. Parent Supervising/Directing

Supervising/Directing involved parents directly instructing their children. In this interaction, parents were the lead and children did not play an agentic role, but instead listened to and followed the parents' instructions. In the context of the coding game, we observed parents engaging in Supervising/Directing when they directly told their child what to do to solve the problem. Supervising/Directing generally led to children's

engagement in Algorithm and Procedure and Debugging, though some instances of Data Collection also occurred. When parents used this role to help children work through their algorithms, children exactly followed their parents' dictated moves to create an algorithm or to fix their algorithms. In simply following their parent's directions, children did not fully engage in computational thinking. In one case, a child initially followed a parent's dictated instructions, which failed to solve the level, then proceeded to independently solve the level without much trouble. Therefore, Supervising/Directing during algorithm generation and debugging may keep children from thinking independently generating solutions.

We observed that parents used Supervising/Directing either when they had just started the activity or when children appeared to be struggling with an aspect of the activity. For example, as they started doing the activity, some parents would read the task to themselves, describe the task and then tell the child how to code the robot with specific moves. Another common example was that if a child struggled with entering the path for the robot, a parent may have listed the necessary moves, allowing the child to simply press the buttons for the algorithm generated by the parent. The child engaged in the Algorithms and Procedures competency by following an algorithm, but their engagement did not allow them to participate in generating the algorithm, which Fagundes and colleagues [38] suggest is a more advanced form of engaging in the Algorithms and Procedure competency. In Debugging, parents told their child exactly how to solve an error in the algorithm by telling them how many code moves to delete and how many to add, and in most instances, the error was identified by the parents themselves. Since the child was not a participant in the process of generating and debugging the solution themselves, their engagement in the necessary competencies is limited to just "fixing" the error rather than also diagnosing and explaining the error [61].

### 5.2.2. Parent Facilitation

When engaging in Facilitation, parents took actions to reduce the challenge for a portion of the activity for their children without directly instructing them. While facilitating, a parent might prompt their child to think in a certain way by saying "Why don't you look at that again", or "Okay, let's move to the cat next". In this type of interaction, parents allowed their children to retain agency and engagement in the activity and competencies. During Facilitation, parents led the interactions but also gave their children the chance to think and work independently, usually by providing some information to help their children solve a problem while withholding the exact solution.

This child–parent interaction primarily resulted in children's engagement in Debugging and Algorithm and Procedure. Parents' Facilitation resulted in children's Debugging in two different ways. In the first case, parents identified the error/problem for children and used Facilitation strategies to have the child address the error. Instances included parents encouraging their children to fix the error without giving them further information by asking questions, such as "let's fix this code. How do you think you can fix it? Can you think of a way you can change this code? What move do you think you need to use here so the robot doesn't hit the wall again? What needs to be deleted here?"

In the second case, parents did not identify or address the error for children, but taught them strategies they could use to find and address the error. Instances included parents noticing an error in a child's code before it was run, and using phrases such as "I think you should check that again, you may want to use fingers to trace [the] robot's movement before you hit run". This allowed the child to work through his solution independently while practicing a strategy that they could use later. As a result, Facilitation allowed children to engage in the CT competencies, but not as fully as letting the children lead the interaction on their own.

### 5.2.3. Child–Parent Co-Learning

In Co-learning, neither parents nor children prompted or instructed the other. They worked together on the activity, sharing information and insights, or they worked together

in parallel. Co-learning was similar to Facilitation, as we generally observed Co-learning when parents took an action to reduce the challenge of a problem for their child. However, in Co-learning, parents did not encourage their children to approach the problem in a specific way. Instead, in Co-learning, parents helped their children with the problem at hand without attempting to lead the interaction. As an example, when one family began the game's first full level, a parent remarked "Oh, this is basically that over there", referring to the portion of the exhibit containing the physical play space. This play space had the same maze layout as the first level of the game. By providing relevant information to the child without explicit direction, the parent acted as an equal participant in the game. This child–parent interaction resulted in the child's engagement in Algorithm and Procedure, Debugging, and Abstraction. The most prevalent example occurred in the generation of an algorithm for a level. In many cases, children struggled to keep track of where the robot would be at any particular point while writing their code. Frequently, parents would help by simply holding a finger to the screen to identify the robot's position. Then, the child could enter the next instruction confident in the robot's position at that point. Therefore, in Co-learning, the child generated the solution on their own (Algorithm & Procedure), and the parent simply helped them with the specific implementation of their solution. In some instances, parents used the same strategy during the enactment of Debugging. When a child identified the problem and was trying to address the problem, the parent would hold their fingers on the screen or hold their child's fingers on the screen and trace the codes written by the child, which helped the child find ways to address the code. In some cases, parents would also share information to help children engage in abstraction. Generally, parents would help children identify real-world counterparts to the items in the game scene, such as stating "Oh, these are walls!" in reference to the blue lines of the maze. This process of sharing information while allowing children to act independently allowed children to engage thoroughly in the different competencies without getting lost or overwhelmed.

### 5.2.4. Parent Becoming a Student of the Child

When parents interacted with the child as if they were a student of the child, they signaled to their child to take more control of the activity. Parents interacted with the child by asking open-ended questions, such as "I'm not sure how to do this, do you?" and "What do you think we should do next?". These questions encouraged the child to lead the activity, try to create an algorithm that works, to find and fix the error, and to undertake more of the problem-solving process on their own, while their parent remaining engaged in case they needed help.

This interaction resulted in the child's engagement in Abstraction, Problem Decomposition, and Debugging. Some parents asked children about the nature of different gameplay elements, such as "What do you think these blue things are?", referencing the walls making up the maze. This question led children to abstract the different objects in the game's environment as obstacles that the robot could not pass through. Other parents would ask questions such as "What do we need to do next?", implicitly guiding the child to break a level into a series of challenges and enabling children to engage in problem decomposition, only considering the next small portion of the level. When a parent acted as a student of the child in Debugging, they usually asked open-ended questions that left an opportunity for debugging, but allowed the child to identify and address issues. In one instance, a parent noticed that the child's route worked but did not follow the best path through the level, so they asked, "Do you think we found the fastest way?" This question prompted the child to consider their path and engage in debugging to optimize their solution to the level. Open-ended questions challenged the child to think through the problem independently and allowed them to engage deeply in the relevant competencies.

### 5.2.5. Parent Disengagement

Disengagement occurred when a parent fully disengaged from all or part of the activity. Parents may have said "You try this one on your own", or "Okay, your turn". Disengage-

ment was also usually accompanied by physical separation; when parents disengaged, they usually took several steps back from the activity's display or they engaged in other activities. Very few parents disengaged during the activity, but parents who did disengage could be categorized into two groups: (1) parents who purposefully disengaged at certain times to let their children be independent, and (2) parents who disengaged consistently throughout their time at the exhibit. In both groups, children continued to engage in the activity and relevant CT competencies after their parents disengaged. In the first group, children were left alone to lead the activity themselves, as they had already showed evidence of being capable of solving the problem themselves, particularly generating algorithms and debugging when needed. However, most of the children in the second group found progressively fewer optimal solutions to the levels of the game the longer their parents disengaged. One possible reason could be that children in the second group did not build the conceptual foundation of the game, given that they did not receive enough initial support from parents during the first level(s). Additionally, we noticed children sometimes became distressed at their parents' disengagement. This distress persisted even when children completed levels independently.

### 5.2.6. Parent Encouragement

Parents engaged in encouragement when they praised or reassured their child during or after the activity. Many parents encouraged their children after they completed a level, generated an algorithm or found and fixed an error within their codes. Encouragement happened by using phrases like "Good job! You did it!" or "Awesome! You found the best answer!" While encouragement did not necessarily directly result in any computational thinking competency, we believe that parental encouragement was helpful in maintaining children's engagement in the activity.

## 6. Discussion

The findings of this study highlight that both parent–child and child–computer interactions were necessary to enable CT thinking for children. Child–computer interactions made the occurrence of CT possible for children, and child–parent interactions involved the parents prompting and assisting their children in engaging in those competencies. For example, for the Algorithms and Procedures competency, the map helped them to identify what algorithm should be created, and the buttons enabled children to enter the algorithm, but parents gave their children the information and prompting necessary for them to execute the competency. Similarly, in Debugging, the computer provided tools that enabled children to identify when their code had an issue, but without the help of parents, many children would not have been able to locate the position of the error in their code or eliminate it. In one case, a child entered a set of moves for the robot, and a parent noticed an error before the family ran the code. The parent initiated the child's engagement in Debugging by identifying and pointing out the error to the child, and then the child used the computer to complete the debugging. The child read back the recorded moves in the code log while "simulating" the robot by tracing the moves on the map. Therefore, the computer provided the child with the capability to engage in computational thinking, while the parent helped prompt the computational thinking engagement. Certainly, these children could have used the buttons on their own to enter or delete moves for the robot, but they may not have understood how to enter the moves in an organized way to complete the task without their parents' assistance. Many of the other competencies shared this balanced need for child–computer interactions and parent–child interactions. In addition to encouraging children to use competencies enabled by the coding game, parents played a significant role in communicating the context and purpose of the activity to children who would otherwise not have been able to read and understand the instructions. Further, without parental engagement, some children could not have completed the various challenges throughout the activity. Since the game featured the same five levels for all participants,

parents could "select" the difficulty of the activity for their children by taking more or less control over the activity.

The findings of this study are consistent with the results of other studies conducted at a similar time, such as one that demonstrated that child–parent interactions were necessary when children used coding kits [57,62]. Moreover, we believe that including parents in CT activities facilitates the necessary social interactions as principles of facilitating constructivism-based learning suggested by Papavlasopoulou, Giannakos, and Jaccheri [63]. However, the findings also add to the literature by providing examples of ways those parental interactions facilitate different CT competencies. For example, we see that Algorithm and Procedures was practiced by children when parents supervised, facilitated, or engaged in co-learning with children. Additionally, the findings of this study add to previous research that has shown that computer scaffolding is necessary and helpful for children's engagement in CT [16]. Similarly to previous studies [64,65], the different components of the computer-based coding game provided feedback to these young learners, which resulted in the active engagement of children in CT competencies. Some types of feedback, such as the "Oops" sign, or the robot moving back to the beginning, motivated children's engagement in Debugging. Thus, while we understand some types of computer-given feedback may result in passive learning, as suggested by Lye and Kuh [16], we believe appropriate reinforcement of feedback by the coding game could provide motivation for children to practice CT competencies.

### 6.1. Implications for Designers

While the exhibit designers chose to design the computer-based coding game and the overall exhibit in a way that prompted and depended on parent–child interactions, unfortunately, in a real science-center environment, not all parents will dedicate the same concerted engagement with a single activity as they did in the observations captured for this study. As discussed, without parental input, many children would have struggled to engage in the CT competencies encouraged by the exhibit. While many components of this coding game were developmentally appropriate, several improvements could be made to the computer-based coding game to account for instances of children playing alone or with little assistance from and engagement by their parents.

Use imagery and auditory feedback. The most important improvement that could be made to the computer interface is the method by which information is conveyed to children. Feedback in the form of text displayed on the screen was one of the main sources of feedback that designers used in this exhibit. This feedback was challenging to understand for many of the observed children who could not read well or at all. To improve the relevance of this feedback, imagery and auditory cues should be used instead of or alongside the written feedback. As an example, children responded very well to the "OOPS!" bubble, which was jagged and brightly colored. In this case, the visual effect added significance to the feedback beyond simply displaying the word "OOPS" somewhere on the screen. Lee and colleagues [66] argue that programming interfaces allow designers to utilize a variety of tools to provide tips to the learners and highlight the critical information they need. Thus, when designing for young learners, we encourage other designers to consider the use of visual guidance and/or auditory guidance to make the interaction more meaningful and effective for children.

Longer display of feedback. In addition to improving written feedback, we noticed a need to display feedback on the screen for a longer time to allow the children to read and comprehend the feedback. During the hands-on and tangible coding games, children could spend as much time as needed to create their algorithm. However, the time may be limited in computer-based coding. Increasing the time that feedback is displayed on the screen is especially important for younger children, as their information processing abilities are developing [67]. In this coding activity, some of the tools meant to allow for debugging could be improved simply by increasing the amount of time they are displayed. In the case of the medicine bar, the circles showing which animals had received medicine were only

filled during the time the robot was moving; if the robot crashed or the run ended, the bubbles were immediately emptied. Instead, the medicine bar circles could remain filled after a run was attempted, so that players could see which animals they missed. Further, a tool to enable children to run the movement path slowly or a single step at a time could be helpful. This would allow children to identify exactly the place in their code that contained an error. These tools would reduce the need for children to pay very close attention to a fast-moving robot and instead help them develop the ability to debug in a meaningful way.

Gradually expanding the coding complexity. More complicated movement and coding capabilities could improve the relevance of the game to more advanced children. Programming and coding activities for school-aged learners should be constructivism-based problem-solving [18], where children can construct understandings of CT through social interactions with parents, other adults, or peers, paired with scaffolding and opportunities for reflection so children can grow in the zone of proximal development [39]. Given the nature of this activity, children in this study gradually developed the CT competencies and moved to higher levels of the coding game. However, some of the children, especially those who were 7 years old (in second grade), may not have been challenged enough; they easily interacted with different features of coding game, were able to read the prompts, and mastered the higher challenges quickly. Introducing turning instead of simple four-directional movement would be a simple step to increase the challenge of moving the robot through the maze. Loops, sensors, and rudimentary logic could allow children to create algorithms that allowed the robot to autonomously navigate through any maze. These algorithms are certainly more difficult to grasp, but previous studies have shown that children as young as first grade are able to create complex algorithms [68–70]. Additionally, as the game becomes more complex, the coding game should provide space for children to be able to debug while coding. In the game's current implementation, moves in the middle of a sequence cannot be edited without deleting all of the moves entered after them. If a move in a long sequence is entered incorrectly, all of the moves afterward must be re-entered, even if they were correct. Implementing a cursor to select individual moves in a string could enable much easier debugging.

### 6.2. Implications for Educators and Facilitators

We captured diverse child–parent interactions, ranging from parents directing the activities to children leading independently. These interactions enabled children's engagement in CT competencies in different ways. For example, considering Algorithm/Procedures and Debugging, Facilitation and Co-learning helped children to fully engage in creating an algorithm, testing it and then identifying and fixing the bug (error) in the algorithm, while supervision only allowed children to follow their parents' algorithms and fix errors in the way parents suggested. The findings of earlier studies that examined collaboration between parents and children using coding kits [71,72] similarly highlighted that when collaboration occurs in forms that allow children some freedom, such as co-learning and parents' facilitation, children can more easily develop CT competencies. Thus, we encourage practitioners to provide the needed guidance and scaffolding for children, but also allow them to lead the activity.

We observed in this study that when parents purposefully disengaged from the activity, children engaged in CT. They created algorithms using simulation and they practiced debugging on their own. In previous studies, evidence of parents' disengagement was observed as a play pattern between preschoolers and parents during a digital game. Hiniker and colleagues [72] considered parents in this play pattern to be acting as "bystanders" where they disengaged and left children alone to play with their tablet. In Hininker's study [72], no evidence of direct learning was observed as a result of this play pattern. However, as we closely analyzed parents' disengagement in this study, we realized that it usually happened after parents had already provided enough facilitation and direction to their children. Therefore, in this study, disengagement still resulted in children's CT.

We encourage practitioners to employ disengagement when children have already shown evidence of being able to practice CT competencies.

While these interactions provide different amounts of freedom for children to make decisions and lead the activity, we are not suggesting one is better than the other. We believe each of these interaction types may be necessary depending on the child and the specific situation. Supervising limits the child's freedom to make decisions; however, it seems to be necessary at the very beginning of the activity as the child is trying to learn how each component of the game works. Previous studies have similarly discussed the role of adults as teachers at the very beginning of coding activities, where adults would directly teach school-aged children how to engage in coding activities and create and debug codes [71–73]. On the other hand, as the child develops proficiency in CT competencies over time, disengagement seems to be the appropriate interaction approach for parents. Previous studies have observed that children enjoy teaching skills to others, and this is a motivational strategy for them to develop coding skills [66]. In this study, we similarly observed that the parent's becoming a "student of the child" resulted in effective CT engagement by children.

We also acknowledge that different children may need more time to develop coding-related skills, and therefore they may benefit from having more directed guidance or facilitating questions even at the more advanced levels. Diversity in interactions has been seen in previous research [57] and is also consistent with sociocultural and constructivist theories of learning, as children have different needs and require different types of support by a more knowledgeable other to engage in learning and construct knowledge to reach their zone of proximal development [39].

## 7. Conclusions

In this study, we found evidence that 5- to 7-year-old children are capable of engaging in CT competencies, specifically Algorithms and Procedures, Debugging/Troubleshooting, Abstraction, Data Collection, and Problem Decomposition. We also found that both the computer-based coding game and the children's parents supported the children's engagement in CT, and that with this age group in particular, it was important to consider the whole system of child–computer–parent interactions, rather than looking only at child–computer interactions.

In this study, we saw that parents used a variety of more directive and less directive approaches as the children gained familiarity with the game and experience with engaging in CT competencies. Overall, we noted that it was helpful for children's engagement in CT competencies when parents provided more support and direction initially, especially regarding introducing the activity and helping children develop CT competencies. However, parents shifted to a less hands-on role where they allowed children to lead the activity and practice those competencies on their own. This finding is particularly important, as parents (and facilitators in some informal learning environments) have a unique educational position. Unlike teachers, they do not always focus on providing directions and instructions on games, including coding and programming.

*Limitations and Future Research*

While we have conducted a systematic data analysis with three researchers involved, we acknowledge that all research projects may carry limitations, and researchers' prior experiences impact the ways we focus our analyses and interpret our findings. In our video findings, there may have been critical moments that we have missed in analysis. However, we minimized the potential for a single researchers' perspective to bias the analysis and the likelihood of missing critical moments in the video recordings; this was achieved through the inclusion of three different researchers who had three different sets of prior professional and personal experiences.

The families who participated in this study were small groups of parents and children. While we believe, as stated in implications, that children are different and may need

different interventions, we do believe that the findings of this study may be applicable to other learning settings, including home and afterschool programs where parents and practitioners facilitate the engagement of small groups (i.e., family-sized groups) of young learners in computational thinking competencies. We believe that practitioners and parents should adapt different interaction strategies and combinations of strategies for different children based on their needs to engage in different CT competencies. However, future research is needed to examine ways these interactions could be used with larger groups of learners and in more formal settings. Future research should also focus on learners with different age and different abilities and examine the effectiveness of these interactions in different coding activities.

Future research should similarly consider the social contexts of child–computer interactions, as parents and other family members are often involved in child–computer interactions. In this study, we only focused on the interactions of individual children as they interacted with a parent and the coding game, and we did not examine how siblings interacted with each other. Gaining insight into the ways children interact with each other while engaging with CT activities can further enrich our understanding of how children learn CT and learn to code, as we consider child–child–computer interactions. Thus future research should consider examining the interactions of young children and their siblings and peers to further refine our understanding of CT learning for early learners.

**Author Contributions:** Conceptualization, M.E.C. and H.E.; methodology, H.E. and M.E.C.; formal analysis C.O. and H.E.; investigation, H.E. and C.O.; resources, M.E.C., data curation, H.E. and C.O.; writing—original draft preparation: H.E. and C.O.; writing—review and edit, M.E.C., H.E. and C.O.; supervision, M.E.C.; project administration, M.E.C.; funding acquisition, M.E.C. All authors have read and agreed to the published version of the manuscript.

**Funding:** This material is based upon work supported by the National Science Foundation under Grant No. DRL-1543175). Any opinions, findings, and conclusions or recommendations expressed in this material are those of the author(s) and do not necessarily reflect the views of the National Science Foundation.

**Institutional Review Board Statement:** The study was conducted in accordance with the Declaration of Helsinki, and approved by the Institutional Review Board (or Ethics Committee) of Purdue University (protocol # #1507016230 approved 7 October 2015).

**Informed Consent Statement:** Informed consent was obtained from all subjects involved in the study.

**Data Availability Statement:** The data source for this study was video recordings of the families. These are not available due to privacy and ethical restrictions.

**Acknowledgments:** The computer-based coding game and exhibit that was a focus of this study was designed in collaboration with the Purdue Exhibit Design Center. We could not have conducted this study without this resource. We are also grateful for our partnership with the science center that allowed us to use their space to conduct this study. Finally, this study was part of a larger project with a larger team of faculty, staff and student researchers from Purdue's INSPIRE Research Institute for Pre-College Engineering.

**Conflicts of Interest:** The authors declare no conflict of interest.

## Appendix A

The INSPIRE CT Framework was used as a coding scheme for capturing children's computational thinking in this study and several previous publications written by INSPIRE researchers [12,21,35].

| CT Competency | INSPIRE Definitions | Examples |
|---|---|---|
| Abstraction | Identifying and utilizing the structure of concepts/main ideas | "Oh, this [the OOPS sign] means we did it wrong." |
| Algorithms and Procedures | Following, identifying, using, and creating an ordered set of instructions (i.e., through selection, iteration and recursion) | "The students are interacting with the coding game. One student enters some codes while using his fingers as he counts how many moves the robot must take, then he hits Go. The code does not work, and he says: my bad. The second student deletes all the codes that were written. Then he starts from the beginning and creates new codes." |
| Automation | Assigning appropriate set of tasks to be done repetitively by computers | No opportunities for automation are provided in this activity. |
| Use of Data | Any use of data, including data collection (gathering information to solve the problem, analysis (making sense of data) and representation (organizing and depicting data) | "Child started counting the blocks to learn how many moves the robot has to take to get to the rabbit." |
| Debugging/Troubleshooting | Identifying and addressing problems that inhibit progress toward task completion | Child sees the sign Oops on the screen and realizes an error in the codes, by saying "oh, my bad". He then deletes all the codes and starts from the scratch and creates new codes. |
| Pattern Recognition | Observing patterns, trends and regularities in data (Google) or creating patterns | Child recognizes the patterns for creating the codes that can move the robots to the top of the maze. |
| Problem Decomposition | Breaking down data, processes or problems into smaller and more manageable components to solve a problem | Mother reads the instruction and tells the child that the robots need to get to the cat, rabbit and the dog in no specific order. The child says, "I'm trying the rabbit first, then the dog and then rabbit". She then enters the codes to follow the order she planned. |
| Parallelization | Simultaneously processing smaller tasks to more efficiently reach a goal | No opportunities for parallelization are provided in this activity. |
| Simulations | Imitating natural or artificial processes by using a developed model or representation. | The child enters some codes to take the robot to the cat. Before he runs the code, he uses his fingers and traces the path he expects the robot to pass. |

**Appendix B**

This codebook was used for the child–parent interactions. This table was previously published in Ohland et al. [57].

| Role | Freedom | Definition | General Examples |
|---|---|---|---|
| Supervising/Directing [2–4] | Most adult-led | Parent directly instructs child to act in a specific way. | "You guys do the same path in there. So you've gotta go to your right" |
| Facilitation [3,4] | Adult-led | Parent makes suggestions and prompts the child to think in a specific way. | "Do you not think that would have been quicker if you went to red first?" |
| Co-learning [4] | No leader | Both parent and child work together on a task together; neither is the leader and no prompting occurs. Parent and child share information with each other. | Parent shows the current location of the robot in the game while the child enters instructions: "Oh, these are walls!" |
| Student of the Child [3,4] | Child-led | Parent prompts the child to take the lead in the activity. | "So we know that you got the cat. Now what?" |
| Disengaged | Most child-led | The parent completely disengages from the activity, leaving the child to continue on their own. | "The mom says 'Do what you want.' and she steps back" |
| Encouragement [4] | Ancillary | Parent reassures or encourages the child while they are working on a task or after they complete a task. | "Awesome!" "You found the best answer" |

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
