# Peer review of "Characterizing Child–Computer–Parent Interactions during a Computer-Based Coding Game for 5- to 7-Year-Olds"

_education, doi:10.3390/educsci13020164_

Round 1

Reviewer 1 Report

The paper has a clear focus: to characterize children’s interactions with parents and the coding game, and how these interactions support and enable children’s engagement in different  CT competencies. The motivation behind the research objectives was sufficiently described in the introduction, but the rationale section can still be improved. While it was apparent that there is merit in investigating the children-computer interactions and how they enable the engagement in CT competencies, the rationale for children-parent interactions is less elaborated. It is important to provide stronger rationale for the latter because it is less intuitive compared to the former, not to mention, interesting. The paper also presents an appropriate theoretical framework (Social Constructivism) that enlightens the interpretation of the results. Furthermore, the tenets of the theoretical framework were thoughtfully incorporated in the conclusion and implications section of the paper. The study has a sound methodology, and the data analysis and coding assignment are reasonable. The results section is comprehensive and well-organized especially given the qualitative nature of the work. Finally, the conclusion section does a good job of summarizing the findings and presenting the implications for parties of interest such as designers and educators/facilitators. However, the conclusion section lacks a limitations section which could have tempered if not supported the implications stated in that part of the paper.

While the paper has the essential elements of a publishable manuscript, I believe that it can still be improved, based on the comments above. Furthermore, here are some VERY minor typographical details that I caught:

Inconsistent formatting for lines 51-54, line 185, and lines 227-233

Line 246: insert appropriate space in “ 5- to7-yearold children”

Line 298: insert comma between “analysis” and “we”

Line 326: insert comma between “1” and “we”

Line 368: insert space between “approximately” and “1”

Line 380: The paragraph break is odd. Three competencies were described in the previous paragraph. Abstraction was part of the previous paragraph, but then the other two competencies were described in a separate paragraph.

Inconsistent paragraph spacing in pages 11-14, especially line 449

Line 902: "S." instead of authors full first name. "Basic Books" is also highlighted in gray

Line 932: formatting

Line 996: formatting of "[49]"

Line 1034: entry [68] is not in a separate line

Author Response

Thanks for providing insightful feedback.

We have made the rational stronger by backing it up with more literature and discussions on the. 

We have added a separate section for  Limitation and Future research under the conclusion. We discuss possible limitations of our study and also needed research for future. 

Thanks for capturing where we missed formatting the paper. We have addressed all the minor edits you have captured. 

Reviewer 2 Report

This was an interesting study of child-computer interactions and the child-parent interactions that promote computational thinking in young children. In my opinion, the research is well-grounded. I only have to make three suggestions to the authors to improve the quality of the paper.

1. I recommend the authors structure the discussion section into two subsections according to their research questions. The first research question should also be discussed to the same degree as the second one. 

2. Abstract is a very important section of a paper. Please consider reducing its size, a big abstract is not necessarily a good one. Therefore, I will suggest using the first sentence to identify the research area and the second sentence to underline the research gap. The rest of the abstract should use one sentence to selectively present each of the following: methodology, results, and implications. 

3. The data analysis section could be improved by using a figure to present the various steps of the analysis clearly.

Author Response

Thanks for providing us with insightful feedback. 

  1. Thanks for your suggestion on the discussion. However, while the findings section is nicely divided by the research questions, we purposefully combined the two research questions, as we saw the intertwined connection of both types of interactions with children’s engagement in CT. We have made a slight change in the first sentence, to show that we are talking about both type of interactions combined.
  2. We looked at recent papers published in Education Sciences to find examples of the shorter-style abstract recommended by reviewer 2. However, we noticed that many of the recent papers actually had longer abstracts than our original abstract. Therefore we decided to edit the abstract to improve clarity, but we did not shorten it.
  3. thanks for suggestion on adding visuals for data analysis. We indeed added two figures to clarify the Analysis section.